# Analysis of technical efficiency of irrigated onion (*Allium cepa* L.) production in North Gondar Zone of amhara regional state, Ethiopia

**Tigabu Dagnew Koye** ◉*, **Abebe Dagnew Koye, Zework Aklilu Amsalu**

Department of Agricultural Economics, College of Agriculture and Environmental Science, University of Gondar, Gondar, Ethiopia

* tigabu567@gmail.com

**Data Availability Statement:** All relevant data are within the paper and its Supporting Information files.

## Abstract

Onions are a significant source of revenue and food security for households. Despite their importance in human nutrition, economic benefit, and area coverage, in Ethiopia, onion productivity is significantly lower than it should be. The purpose of this study is to address this gap by examining efficiency variations and determining the variables that affect onion farmers' levels of efficiency in the North Gondar Zone of Ethiopia. The sources of data were both primary and secondary. 205 onion farmers from the Gondar Zuria, Takusa, and Dembia districts were chosen using simple random sampling proportional to sample size. Semi-structured interviews were used to gather primary data from these participants. A Cobb-Douglass production function, a single-stage stochastic frontier model, and descriptive statistics were used to investigate the technical efficiency of onion production at the farm level. The mean technical efficiency of an irrigated onion was 53%, according to the maximum likelihood estimates of the stochastic frontier analysis. By enhancing agricultural methods using current technology, it is possible to raise the average production efficiency of irrigated onions. The stochastic frontier model's maximum likelihood estimates revealed that plot size, Di Ammonium Phosphate, and oxen have a significant effect on onion output; education, livestock holding, experience, and frequency of watering have a positive and significant effect on technical efficiency, whereas family size and marketing training have a negative and significant effect on technical efficiency. Therefore, the government or any relevant bodies should deliver continual scheduled training and an integrated adult education at the existing farmers' training center; modern livestock production techniques; further groundwater resources and proper watering technologies should be used since currently farmers use an inefficient irrigation system, specifically furrow irrigation.

## 1. Introduction

The onion (*Allium cepa*) is an important vegetable crop that is a complementary product to tomatoes and has global commercial importance [1]. Worldwide production of onions in the

**Funding:** This study was funded by the University of Gondar (www.uog.edu.et). The grants were awarded to the first and second authors of this article. The funders had no role in study design, data collection and analysis, the decision to publish, or the preparation of the manuscript.

**Competing interests:** The authors have declared that no competing interests exist.

**Abbreviations:** CC, Contingency Coefficient; CSA, Central Statistical Authority; DAP, Di Ammonium Phosphate; Ln, Natural Logarithm; LR, Log-Likelihood Ratio; MDE, Man Day Equivalent; MLE, Maximum Likelihood Estimator; OLS, Ordinary Least Squares; SFM, Stochastic Frontier Model; TE, Technical Efficiency; TLU, Tropical Livestock Unit; VIF, Variance Inflation Factor.

year 2019 was 99,968,016 Mg, which makes it second to tomatoes and accounted for 9% of the total share of vegetables [2–5]. With 23,907,509 tons, China is the leading onion producer, followed by India with 19,415,425 Mg, Egypt with 3,115,482 Mg, the United States of America with 3,025,700 Mg, Iran with 2,345,768 Mg, and Turkey with 2,120,581 Mg [5–8].

Vegetables in particular, as well as other horticultural crops in general, play a significant role in helping to secure domestic food supplies and generate income [5]. Poor families benefit financially from vegetables because they are valuable food crops that also yield money. According to [9, 10], it is frequently possible to boost the production of a certain vegetable throughout the year when a reliable and economical irrigation system is employed, leading to larger profits. Horticultural farming can be successful when contrasted with the production of main food crops. When cultivable land is scarce, labor is plentiful, and markets are readily accessible, growing fruits and vegetables offers a significant competitive advantage by [9, 11]. According to [12], pepper, kale (Ethiopian cabbage), onion, tomato, chilies, carrot, garlic, and cabbage are some of the most important vegetable crops grown in Ethiopia. According to [9] and, there are opportunities to reduce poverty through the production of horticultural goods, which is often more labor-intensive than the production of major food crops. Therefore, it is feasible to provide additional job opportunities in rural areas with a large labor supply.

Products made from vegetables are advantageous in terms of nutrition, the economy, employment, and social issues [12, 13]. Ethiopian vegetable production and consumption are rising as a result of increasing exports to Djibouti, Somalia, South Sudan, Sudan, the Middle East, and European markets as well as urbanization [12–14]. Exports of cabbage, onions, and chilies increased from 25,300 tons in 2002/03 to 63,140 tons in 2009/10 due to the increasing demand for these products in these nations [12–14].

In Ethiopia, 36.4 million hectares of onions were planted, yielding a total of 273,859 tons [15]. Between 2015 and 2020, onion production increased by 18.7% and onion cultivated area increased by 59.7%, respectively [16]. However, Ethiopia's production of onions (8.89 t/ha) is far lower than the global average (19.32 t/ha) [17], and lower than African countries, with an average value of 10.1 tons/ha [18–20]. Poor agronomic practices, a lack of seeds of improved varieties, diseases and insect pests, insufficient extension services, and high costs of agricultural chemicals are all factors that contribute to Ethiopia's low crop yield [17].

According to [10], onions are one of the most widely produced and commercialized vegetable crops grown under irrigation in the Amhara region. Irrigated agriculture has become more common in the Amhara region in recent years. Within the region, there are around 6,200 small-scale irrigation schemes, with 95% of them being traditional, according to [10]. The irrigation schemes are reportedly held by around 330,000 households (or over 1.9 million people) with an average irrigated land ownership of 0.2 hectares, according to [10]. it is difficult to increase onion production by extending the area of land under cultivation. However, by enhancing existing production technologies, there is an opportunity to increase onion production. Farmers may also be inefficient owing to lack of experience, illiteracy, and other factors. If farmers are proven to be technically inefficient, productivity will be redoubled using current agricultural inputs, agricultural extension services, and available technology. So far as the author's knowledge is concerned, in the study areas, there have been no similar studies on the technical efficiency of onion producers. [20] Review on Economic Efficiency of Vegetable Production in Ethiopia; [21] Analysis of technical efficiency of smallholder tomato producers in Asaita district, Afar National Regional State, Ethiopia; [22] on an evaluation of the efficiency of onion producing farmers in irrigated agriculture: Empirical evidence from Kobo district, Amhara region, Ethiopia; [23] Vegetable Production Efficiency of Smallholders' Farmers in the West Shewa Zone of Oromia National Regional State, Ethiopia. In order to fill the gap, a study was conducted to estimate the technical efficiency of small-scale irrigated onion

producers and to determine the factors that influence their technical efficiency in the North Gondar Zone of the Amhara Regional State of Ethiopia.

### Working Hypothesis

- Smallholder farmers are not technically efficient in irrigated onion production in the study area.

- Socioeconomic variables do not significantly influence technical inefficiency

## 2. Materials and methods

### 2.1. Ethics statement

To care for both the study participants and the researchers, ethical clearance letters were obtained from the University of Gondar Research office and the districts of Chilga, Dembia, Gondar Zuria, and Takusa. During the survey, official letters were written for each kebele, informed verbal consent was obtained from each client, and confidentiality was maintained by giving codes to each respondent rather than recording their name. Clients were informed that they had the complete right to discontinue or refuse participation in the study. As a result, all participants in the study, including survey households, enumerators, and supervisors, were fully informed of the study's objectives.

### 2.2. Description of the study area

This study was carried out in Ethiopia's Amhara Regional State's North Gondar Zone. It is located 783 kilometers from Addis Abeba and is situated in the northwest of Ethiopia between 11 and 13 north latitudes and 35 and 35 east longitudes. With an average elevation of 2133 meters above sea level, Gondar serves as the zonal capital and is located at $12°35'60.00''N$ latitude and $37°28'0.01''E$ longitudes. About 90% of the labor force in the region is employed in the agriculture sector. In addition to the Tigray region in the north, the Awi zone and West Gojam zone in the south, the Waghimra zone in the east, the South Gondar zone in the southeast, and Sudan in the west, the boundaries are also bordered by these regions. The zone covers 50,970 square kilometers in total [24]. The districts of Takusa, Dembia, and Gondar Zuria were used for the study. Irrigation and rain-fed agriculture have always been used for crop cultivation. Farmers mostly use irrigation to grow vegetables like onions, tomatoes, cabbage, peppers, and potatoes, as well as grains like maize and other related crops, according to the zonal department of agriculture. According to the zonal department of agriculture, onions are provided with a significant amount of the irrigated land allocated and the output volume. This study is illustrated in Fig 1.

### 2.3. Sampling technique and sample size

Respondent producers were selected using a multi-stage sampling technique. Out of all the woredas in the North Gondar Zone, three woredas, Takusa, Dembia, and Gondar Zuria, were purposefully selected for the first stage based on their potential for production (better access to irrigable farms and water, more producers, and higher output levels). Chemera, Chanikie, and Mekonta from the Takusa woreda; Abrjeha and Sufankara from the Dembia woreda; and Sendeba and Ambober from the Gondar Zuria woreda were selected randomly based on their proportion to the total population in the woredas in the second stage. Finally, 205 producers of irrigated onions were selected at random in proportion to the total number of farmers.

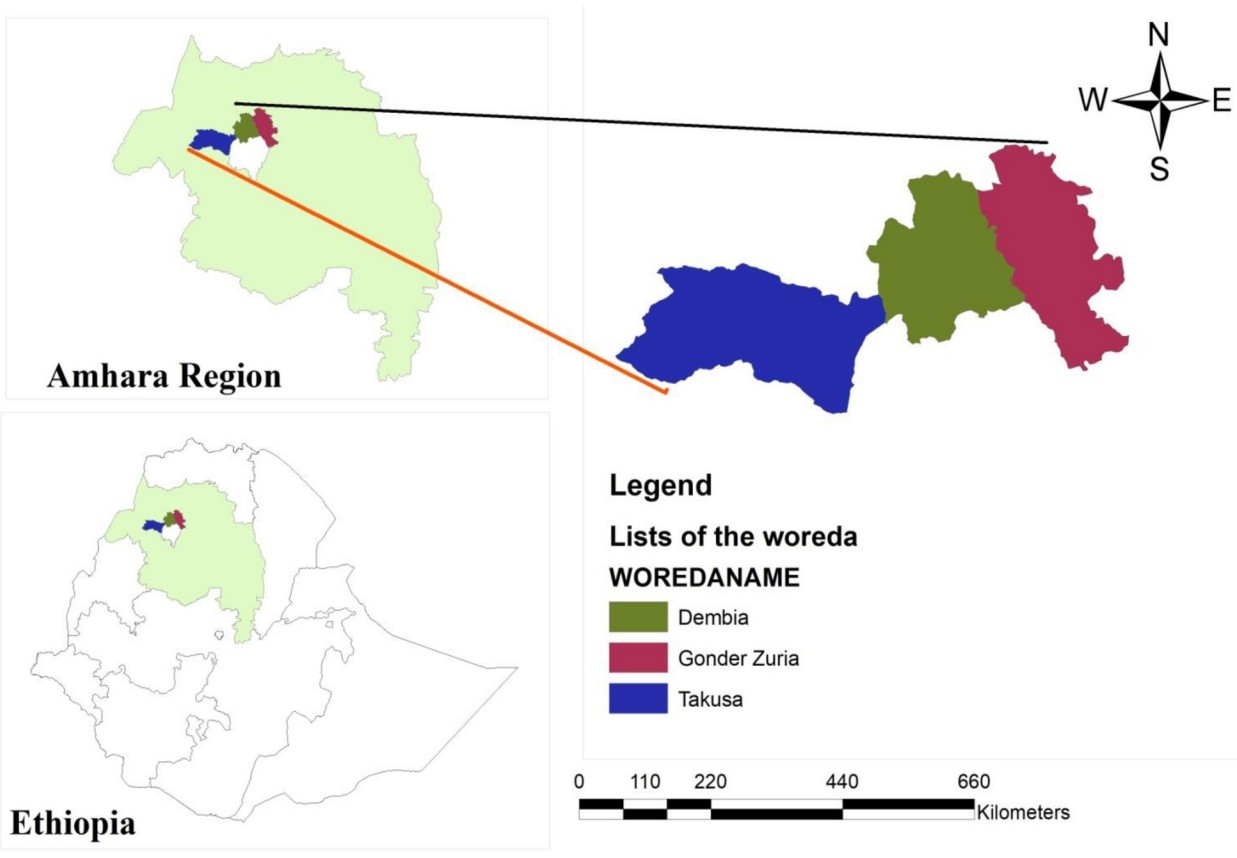

**Fig 1. Map of the study area.** Source: Own developing using Shape file (2022).

## 2.4. Data type, sources, and method of data collection

This study used both primary and secondary data. A semi-structured and pre-tested interview schedule was used to collect primary data during face-to-face interviews conducted by enumerators who had been carefully vetted for recruitment and training. The schedule of the interviews included questions about the farmers' overall output of irrigated onions as well as production related to socioeconomic factors. Secondary data was gathered from a variety of sources, including reports from the Bureau of Agriculture at various levels, non-governmental organizations (NGOs), the CSA, the District administrative office, earlier research findings, and other published and unpublished materials pertinent to the study.

## 2.5. Methods of data analysis and model specifications

Descriptive statistics (such as percentage, frequency, minimum, maximum, mean, and standard deviation) were used to analyze the data. For econometric analysis (namely stochastic frontier), a single-stage stochastic frontier model was used to estimate the level of efficiency and analyze the determinants of technical efficiency of small-scale irrigated onion producers.

The deviation in actual output from the frontier caused by inefficiency and random shocks can be represented using the stochastic frontier approach. Crop production inefficiency is caused by the inefficient use of limited resources. The two main competing techniques for analyzing technical efficiency and its primary determinants are the parametric frontier (stochastic

frontier technique) and the non-parametric frontier (data envelopment analysis) [25]. The non-parametric frontier has come under criticism for failing to consider the potential influence of random shocks like measurement errors and other noises in the data [26].

The stochastic frontier model was chosen because it is suitable for analyzing farm-level data where measurement errors are considerable and the weather (natural hazards, unexpected climatic conditions, pests, and diseases) is likely to have a significant impact [27, 28]. The cost, profit, or production relationship among inputs, outputs, and environmental factors is specified using a functional form and this approach—also known as the econometric frontier approach—allows for random errors [29]. [30, 31] established the Stochastic Frontier Approach (SFA) independently. Based on specific distributional assumptions for the technical and economic inefficiency scores, SFA can be extended to quantify inefficiencies in individual production units [21, 32].

The most typical functional forms are the Cobb-Douglas and transcendental logarithmic (translog) functions. Several functional forms have been established to examine the physical relationship between input and output [33]. Although the Cobb-Douglas is less flexible and simpler, it satisfies the condition of being self-dual, allowing the study of allocative and economic efficiency [34]. On the other hand, the Cobb-Douglas functional form makes significant assumptions about the nature of agricultural technology (it assumes constant elasticity over the input-output curve and unitary elasticity of factor substitution) [11].

The trans-log production function, as opposed to the Cobb-Douglas production function, is a more flexible functional form since it allows for parameter non-linearity and takes into account variable interactions. It does, however, have several limitations. Due to the degrees of freedom, it does not produce coefficients with suitable sign and magnitude, and multicollinearity among the explanatory variables is typically present when estimating a trans-log production function [34]. According to the aforementioned literature review, neither of them can be deemed superior to the other without the use of a test of hypothesis. In this study, a test of the hypothesis was used to choose one of them over the other. The model specification of the stochastic production frontier may be tested using the likelihood ratio test (LR Test) [33]. The LR Test's null hypothesis is that all interactions and second-order terms related to the trans-log specification are equal to zero [35].

Numerous empirical studies have largely used the Cobb-Douglas model, particularly those focusing on farm efficiency in developing nations [36]. It is written as follows:

$$Y_i = F(X_i \beta) \exp(v_i - u_i) \quad i = 1, 2, 3, \ldots, 205 \tag{1}$$

Where: $Y_i$ = Onion output, i = the $i^{th}$ farmer in the sample, $X_i$ = a vector of inputs used by the $i^{th}$ farmer, $\beta$ = a vector of unknown parameters, $V_i$ = a random variable which is assumed to be normally and independently distributed, and $U_i$ = farm-specific technical inefficiency in production and nonnegative random variable. The Cobb–Douglas form of stochastic frontier production is stated as:

$$\ln Y = \beta_0 + \sum_{j=1}^{6} \beta_j \ln X_{ij} + v_i - u_i \tag{2}$$

Where: ln = natural logarithm, $X_{ij}$ = is the quantity of input j used in the production process including oxen (ODE), labor (MDE), plot size (ha), DAP (kg), Urea (kg), and seed (kg). After estimating the technical inefficiency ($u_i$) from Eq (2), the technical inefficiency model was

specified as:

$$\mu_i = \delta_0 + \delta_{1i}(\text{Age}_i) + \delta_{2i}(\text{Educ}_i) + \delta_{3i}(\text{Famsze}_i) + \delta_{4i}(\text{TLU}_i) + \delta_{5i}(\text{Expr}_i) + \delta_{6i}(\text{Frqcnt}_i)$$
$$+ \delta_{7i}(\text{Slope}_i) + \delta_{8i}(\text{Trngprdn}_i) + \delta_{9i}(\text{Trngmkt}_i) + \delta_{10i}(\text{wateringfrnq}_i) \tag{3}$$

Where: The subscript i, indicates the $i^{th}$ household in the sample; $\delta_0, \delta_{1i}, \ldots, \delta_{10i}$ are parameters to be estimated.

Using the available technology, the farm-specific technical efficiency is defined as the ratio of observed output ($Y_i$) to the corresponding frontier output ($Y_i^*$), which was specified as:

$$\text{TE}_i = \frac{Y_i}{Y_i^*} = \frac{E(Y_i/U_i, X_i)}{E(Y_i/U_i = 0, X_i)} = E[\exp(-U_i)/\varepsilon_i] \tag{4}$$

TE takes the value on the interval (0, 1), where 1 indicates a fully efficient farm.

By using the Frontier 4.1 [37] computer program, where the variance parameters are stated in terms of, the maximum likelihood estimates for the stochastic frontier parameters are obtained.

$$\sigma^2 = \delta_v^2 + \delta_\mu^2 \quad \text{and} \tag{5}$$

$$\gamma = \sigma_\mu^2 \Big/ (\sigma_v^2 + \sigma_\mu^2) \tag{6}$$

Where: $\sigma^2$ it is the total variance of the model and the term $\gamma$ represents the ratio of the variance of inefficiency's error term to the total variance of the two error terms defined above. The value of variance parameter $\gamma$ ranges between 0 and 1.

## 2.6. Variable definitions and expected signs

Input-output factors, as well as technical efficiency in relation to demographic and socioeconomic characteristics, are both considered in technical efficiency analysis.

**The dependent variable** was the total physical output of onions for the production year, expressed in kilograms (first using local measurement, then translating it to the scientific measuring unit).

The following definitions were given for independent variables, which include inputs used in the production of onions and socioeconomic factors affecting technical efficiency:

**Plot size** is the total area of the irrigated plot in ha allocated by the $i^{th}$ household for the production of onions. The difference in farming efficiency among farmers in the study area is supposed to be found. It's critical to assess whether or not comparatively large farms are more efficient than smaller ones. Manageability may decline as a farmer's farm size rises. A positive coefficient is expected since land is the primary factor of production [38].

**Fertilizer** is the total amount of fertilizer (DAP and urea) used by the $i^{th}$ household to produce irrigated onions, measured in kilograms. The amount of fertilizer used was anticipated to increase output, but an overdose can result in a subpar yield or complete crop failure. As a result, the use of chemical fertilizers will boost the level of production and the efficiency level of the farmer will be positive [39].

**Oxen power** is the total number of oxen days utilized by the household in order to prepare the soil for the growth and transplantation of onion seedlings. The number of oxen power utilized per day was taken into account while calculating the amount of draught power used for various farming operations. The farmer can accomplish his tasks quickly and effectively if he has enough pairs of oxen to plow with. Thus, it was hypothesized in this study that farmers' productivity would be higher the more heads of oxen they had.

**Labor** refers to the sum of the work hours put into pre-harvest activities, including applying fertilizer, watering, sowing, transplanting, weeding, hoeing, and cultivation, all of which are measured in terms of man-days. The total weighted labor (Man Equivalent) in person-hours was computed in man-days using a standard conversion factor after adult men and women; child labor, elderly labor, family labor, exchange labor, and employed labor involved in the production process were all registered separately. Given that labor is the primary input in production, a farmer with more labor on hand in the household can promptly employ essential crop husbandry techniques [40].

**Onion seed** is the amount of onion seed utilized in the total by the i^th household, measured in kilograms. Due to intense competition for nutrients, a very high seed density may lead to low onion output, although a very low seed density may also result in low onion output due to under-utilization of the land. Therefore, it was hypothesized that seed quantity affects seed rate, which might have a positive or negative effect on yield.

**Age:** Producers' ages are a good indicator of management quality disparities. Experienced farmers typically experience fewer losses and have greater administrative abilities that they can apply to their production process. It was expected that increasing the number of years spent farming would develop experience, which would improve productivity. Farmers will be able to employ resources in a way that gives the highest possible output as they get older.

**The level of education** is such that farmers who have completed further education are likely to have access to knowledge about the technological production of onions. Furthermore, educated farmers are expected to be more capable of analyzing information and identifying technology that can reduce inputs. It is expected that farmers' education will reduce technical inefficiency.

**Family size (adult equivalent)** is a continuous variable that denotes the size of the household's family. The area's primary source of labor supply is family. The number of people living in a household may increase the farmer's productivity when producing onions. Given that labor is the primary input in production and that the farmer has a large family, timely management of onion plots was expected [41]. Therefore, it was expected that household size would have a positive effect on the farmer's ability to produce.

**Livestock holding:** It refers to the number of livestock owned by the household and measured by the Tropical Livestock Unit (TLU). Livestock possession is perceived as the accumulation of wealth, used for drafting power, manure, financial gain from the sale of butter, and the sale of livestock in times of risk to shop for improved agricultural technologies like seeds, pesticides, etc. Households having outsized livestock can have a better chance to earn more income from livestock. Therefore, the more livestock owned by the farm, the greater the possibility of purchasing improved agricultural inputs and/or the greater the investment in off-farm activity. Again, the farmer also has the chance to get oxen for draught power [11]. As a result, farmers who owned more livestock were expected to be more technically efficient.

**Experience** is a continuous variable that is measured by the number of years the household head spends engaged in farm activity. The number of years of experience is directly related to the farmers' knowledge of onion production. So, it was expected to affect the technical efficiency positively.

**Watering frequency:** Water availability is the main limiting factor of crop productivity over all of the rest due to its paramount importance for normal plant growth and development. As a result, onions are more susceptible to water stress than other crops due to their shallow root systems and the need for frequent irrigation water after a short interval [42]. Therefore, it is assumed farmers who avail irrigation water frequently are expected to affect technical efficiency positively.

**Frequency of extension contact:** Farmers who have better extension contact are expected to be more efficient than others. The more contact the farmer has with the adding service, the

more information and knowledge she or he will have and the better the use of agricultural inputs will be. Therefore, it is assumed that farmers who have frequent contact with development agents are more likely to demand agricultural inputs because of the increased awareness, and it is expected to affect technical efficiency positively.

**Slope:** The slope of the land may affect the level of production. Because steep plots are more prone to erosion and are more likely to be infertile than plain plots, the slopes of the plot were found to be negatively related to technical efficiency [39]. It is a dummy variable that assumes a value of 1 if the slope of the plot is steep and 0 otherwise. It is hypothesized that this will have a negative impact on technical efficiency.

**Training on production and marketing:** Training is an important tool in building the managerial capacity of the household head. Households' heads have gotten those who get training related to crop production and marketing or any related agricultural training is expected to be more efficient than those who did not receive training [43]. It was expected to positively affect the technical efficiency.

## 3. Results and discussions

### 3.1. Descriptive analysis

Table 1 illustrates that households produced an average yield of 2,965 kg/ha of onions, with a standard deviation of 2,958. The household's average Plot size was 0.36 hectares, with a 0.24

**Table 1. Socioeconomic characteristics of onion producers (N = 205).**

| Socioeconomics Variables | | Mean (obs) | STD |
|---|---|---|---|
| Output in (kg) | | 2965 | 2958 |
| DAP (kg) | | 49.8 (195) | 75.8 |
| UREA (kg) | | 33.75 (179) | 24.74 |
| Labor (MDE) | | 62.61 | 55.63 |
| Ox (OXD) | | 7.08 | 5.06 |
| Plot size (ha) | | 0.36 | 0.24 |
| Seed (kg) | | 2.10 | 5.00 |
| Livestock holding | | 7.66 | 4.34 |
| Age (year) | | 44.3 | 9.7 |
| Level of education(year) | | 1.46 | 1.33 |
| Family size (number) | | 6.05 | 2.40 |
| Farming experience (year) | | 5.05 | 3.67 |
| Extension frequency of contact | | 3.43 | 1.91 |
| Watering frequency per 15days | | Number | Percent |
| Two times | | 35 | 17.07 |
| Three times | | 100 | 48.78 |
| Four times | | 68 | 33.17 |
| Five times | | 2 | 0.98 |
| Training on production | Yes | 71 | 34.63 |
| | No | 134 | 65.37 |
| Training on marketing | Yes | 28 | 13.66 |
| | No | 177 | 86.34 |
| Slope | Plain | 163 | 79.51 |
| | Gentle | 42 | 20.49 |

Source: Computed from Field Survey Data, 2015/16

standard deviation. Since the average age of onion producers was 44 years, with a standard deviation of about 10, they are now in the most important part of their employment. The average number of families was found to be 6, with a standard deviation of 2. Small farmland, a large family, and the farmer's management of the production system make it challenging for the farmer to support his or her family. Each extension worker conducted 3.4 visits on average during the production year, which promotes the transfer of new technologies. Training has a significant role in the production and marketing of irrigated onions to increase farmer revenue. For this purpose, almost 65.37% and 86.34% of the family said that they lacked training in production and marketing, respectively. This suggests that differences in technical efficiency among household heads may not be affected by training. Every fifteen days, 17.07%, 48.78%, 33.17%, and 0.98% of the sample families irrigate (water) their onion plots, respectively, twice, three times, four times, and five times. According to the results, irrigation was performed most frequently, three times every fifteen days.

### 3.2. Econometrics analysis

Using maximum likelihood estimation, the stochastic production frontier model's parameters are estimated using the Frontier 4.1 version computer program. Before examining the production frontier parameter estimates and factors influencing the inefficiency of irrigated onion farmers, VIF and CC tests showed that there were no problems with multicollinearity among continuous and discrete variables, respectively (see Tables 1–3 in S1 Appendix).

The first null hypothesis is a test for the presence of the inefficient portion of the combined error term of the stochastic frontier model. This might be used to compare the stochastic frontier model (SFM) used for this study to the traditional average production function (OLS) to see which one best fits the data set. The generalized likelihood-ratio {LR = −2[lnL(H$_0$)−lmL(H$_1$)]} statistic for determining if the frontier has no effect on technical inefficiency is calculated to be {LR = −2*(−228.39903+199.82662) = 57}. This value (57) is greater than the critical $x^2$ (5%, 1) value of 3.84 at the 5% level of significance in Table 2. Therefore, given the appropriate ordinary least squares production function, the null hypothesis was not accepted, showing that the stochastic frontier production function was a sufficient presentation of the data. As a result, the stochastic frontier approach best accounts for the data.

The selection between the Cobb-Douglas and the trans-log production functions as the proper functional form for data was the second null hypothesis that was investigated (see the log-likelihood functional value of the Cobb-Douglas and trans-log production in Tables 3 and 4 in S1 Appendix, respectively). The estimated likelihood ratio will determine which functional form will be used. In Table 2, the computed Log-likelihood Ratio is (LR = −2*(−199.82662+-185.31493) = 29.02), and the critical value of $x^2$ at 21 degrees of freedom and 5% significant level is 32.67. As a result, the null hypothesis that all coefficients of the interaction terms in the trans-log specification are equal to zero was accepted. This means that the data under

**Table 2. Summary of the test of hypothesis.**

| Null hypothesis | Degree of freedom | LR | $x^2$value | Decision |
|---|---|---|---|---|
| H$_0$: γ = 0 | 1 | 57 | 3.84 | Not accepted |
| H$_0$: β$_7$ = . . . = β$_{27}$ = 0 | 21 | 29.02 | 32.67 | Accepted |
| H$_0$: δ$_0$ = . . . = δ$_{10}$ | 10 | 48.87 | 18.31 | Not accepted |

At 5% significance level

Source: Computed from Field Survey Data, 2015/16

examination is accurately represented by the Cobb-Douglas functional form. The Cobb-Douglas functional form was employed to determine the technical efficiency of sample households.

The third null hypothesis explored is that farm-level technical inefficiencies are not affected by farm and socio-economic variables included in the inefficiency model. The inefficiency effect was calculated using the value of the Log-Likelihood function{LR = −2[−224.25967+-199.82662 = 48.87]}. The computed LR value of 48.87 was higher than the critical value of 18.31 at 10 degrees of freedom, indicating that the null hypothesis ($H_0$), which states that all explanatory variables are simultaneously equal to zero, was not accepted at the 5% significant level in Table 2. As a result, these variables also help to explain why sample household efficiency varies.

**3.2.1. Estimation of cobb-douglas's production function.** Table 3 illustrates estimates of the stochastic frontier production function from irrigated onion producers in the North Gondar Zone. A good fit and correctness of the given distributional assumption of the composite error term are indicated by the sigma ($\sigma^2 = 0.60$) which is statistically significant at the 1% level of probability. Gamma ($\gamma$) has a value between 0 and 1. If $\gamma$ is close to 0, it suggests that deviations from the production frontier are entirely caused by random noise, but a $\gamma$ value close to unity suggests that most of the deviations are caused by inefficiency. A gamma value of 0.66,

**Table 3. MLE of parameters of cobb-douglas stochastic production frontier function for onion producers.**

| Variable | Parameter | Maximum likelihood estimate | |
|---|---|---|---|
| | | Coefficient | t-ratio |
| Intercept | $\beta_0$ | 2.3 | 2.9*** |
| LnOx (ODE) | $\beta_1$ | 0.18 | 1.91* |
| Lnlabor (MDE) | $\beta_2$ | -0.06 | -0.89 |
| Lnplot size | $\beta_3$ | 0.64 | 6.4*** |
| LnSeed | $\beta_4$ | 0.05 | 0.93 |
| LnDAP | $\beta_5$ | 0.10 | 1.71* |
| LnUREA | $\beta_6$ | -0.04 | -0.77 |
| Return to scale | | 0.92 | |
| **Inefficiency effect model** | | | |
| Constant | | 0.95 | 1.63 |
| Age | | -0.01 | -0.43 |
| Education | | -0.14 | -1.74* |
| Family size | | 0.12 | 2.74*** |
| Livestock holding | | -0.05 | -1.65* |
| Experience | | -0.08 | -2.09** |
| Extension frequency | | 0.05 | 0.20 |
| Slope | | 0.08 | 0.34 |
| Training-production | | -0.36 | -1.17 |
| Training-marketing | | 0.74 | 2.09** |
| Watering frequency | | -0.69 | -2.77*** |
| Sigma-squared | $\sigma^2$ | 0.60 | 4.6*** |
| Gamma | $\Gamma$ | 0.66 | 4.77** |
| LL | | -199.8 | |
| Mean TE | | 53 | |
| Total sample size | N | 205 | |

***,** Represents significance at 1% and 5% probability levels, respectively

Source: Computed from Field Survey Data, 2015/16

which was significant at the 1% level, confirmed the presence of technical inefficiency effects in irrigated onion production in the study area. Thus, it follows that 66% of deviations from the efficient frontier result from sources of technical inefficiency. Or it suggests that 66% of the variation in onion producers' output was related to variations in their technical efficiencies (total variation in output is due to the existence of production inefficiencies). By implication, erroneous data collection and aggregation, adverse weather, the effect of pests and diseases, and similar random factors account for around 34% of the difference in output across producers. Table 3 demonstrated that the coefficients for plot size, oxen power, and DAP all had the expected positive signs, indicating that a unit increase in these inputs will result in an increase in the output of irrigated onions.

The number of **oxen power days** was found to be a significant factor in the output of irrigated onions, with an expected sign and statistical significance at the 10% level. The positive coefficient indicates that a 1% increase in the number of oxen-days used for land preparation will typically result in a 0.18% increase in onion production. Given that plot size and DAP are constant, it is the second crucial factor that influences the volume of onion output. This is in line with [22, 24, 33].

**Onion plot size (ha)** demonstrates the significance of access to the farm plot in explaining the variations in each farmer's output. The elasticity of onion production to farm plots is positive at a 1% level of significance. This indicates that onion output is sensitive to plot size, as increasing onion plot size by 1% can lead to an increase in total onion output of 0.64%. This supports the hypothesis and indicates that farmers with larger plot sizes exhibit significantly better levels of technical efficiency. It is the most crucial input that affects the output of the onion. Given that the amount of DAP and the number of oxen days are constant, it is the critical variable that determines the level of onion output. Thus, improving technical efficiency in onion production in the study areas depends on plot size. Land occupied the highest output elasticity, indicating that it was the main factor of production. This supports the conclusions of [11, 21, 40, 44–50].

The coefficient of the rate of **DAP fertilizer** exhibits an expected positive sign and is statistically significant at the 10% level. Accordingly, the rate of DAP fertilizer must be increased by 1% until it reaches the recommended rate; this will result in an increase in onion production of 0.10%. Given that plot size and oxen days are constant, it is the third crucial variable that determines the quantity of onions produced. This is in line with [33, 51, 52].

A value of <1 return to scale indicates onion farmers were producing at a decreasing return to scale; this is a diseconomy scale of production due to managerial inefficiency in using inputs. Onion production in the study areas was carried out at decreasing returns to scale, as indicated by the coefficient parameters of the summation of the partial elasticity of all inputs that had a significant effect, which was 0.92 in Table 3. Therefore, an increase in all production inputs by 1% will increase onion output by less than 1%. This result corresponds to [50], who found that by increasing all inputs by 1%, the output of white cumin production would increase by less than 1%.

**3.2.2. Determinants of technical efficiency.** The following are the important factors in the technical efficiency of onion producers after estimating technical inefficiency variables using the stochastic frontier model's single-stage estimation approach in Table 3:

At the 1% level, it was revealed that *family size* had a significant negative effect on technical efficiency levels. This was a result of the family's inadequate managing skills in employing the work force that was on hand. This suggests that technical efficiency declines as household size increases. However, this mostly depends on two factors: the number of household members who can work on the farm and the length of time that each member is willing to spend on the household farm. Therefore, the composition and quality of those competent to work on the farm, rather than the fundamental size of the family size, are what matters. This supports the

results of [53–55]. But contrary studies by [25, 40, 56, 57] found that household size actually boosts technical efficiency.

At a 5% level, ***marketing training*** has a negative effect and significantly reduces technical efficiency. This could be as a result of ineffective training, such as training that is not continuous or seasonal, or it might be because of political considerations. This is consistent with [43, 58], but not with [40, 59].

At a 10% level of significance, ***the education level*** of farmers demonstrated a negative relationship with technical inefficiency. A negative sign indicates that more educated farmers are either less inefficient or more efficient in their agricultural production than less educated farmers. This suggests that the technical efficiency of onion production rises along with the level of education. The possible explanation is that individuals with higher levels of education are more qualified for farm management than those who receive short-term training. The latest technological advances and tools are more easily incorporated into farming operations due to education. The production of onions therefore requires special attention. This outcome is consistent with the findings reported by [11, 24, 38, 60–62].

***Livestock holding*** was affected by technical efficiency positively and significantly at 10%. This means that households with more livestock may have fewer problems acquiring inputs like seed, fertilizer, and the like, and that oxen ownership is one of the livestock units taken into account, which helps farmers in land preparation. Thus, an increase in livestock holdings improves onion production's technical efficiency. In terms of livestock holding, the result in this study is congruent with the findings of other empirical works by [50, 63].

***Farming experience*** affected technical efficiency positively and significantly at 5%. It improves household technical efficiency by increasing household agricultural managerial competence through learning by doing [64]. Farming experience has been improving the farmer's skill at onion production. A more experienced farmer's motivation has a lower level of uncertainty about the innovation's performance. Farmers with more experience appeared to have full information and better knowledge and could evaluate the advantages of the innovation considered. The result agrees with the studies of [44, 64–68].

***Watering frequency*** affected technical efficiency positively and significantly at 1%. It is important as per the suggested rate. When it comes to water stress, onions are quite sensitive. Although onions will survive long periods of drought, water handiness is vital for growth and high yields of quality bulbs [69]. Contingent with this, the watering frequency was found to boost technical efficiency through available onion-required water adequately. Additionally, farmers in the study area use the furrow irrigation (flooding) method to irrigate their onion farms.

**3.3. Technical efficiency analysis.** The technical efficiencies of the sampled individual firms are predicted using the maximum likelihood estimates of the Cobb-Douglas stochastic production function coefficients, which are shown in Table 4. According to the findings of the efficiency analysis, the smallholder onion household's technical efficiency ranged from 5.5% to 87.3%, with a mean of 53%. In other words, smallholder households in the study area that produce onions on a small scale typically suffer a 47% output loss as a result of technical inefficiency. This indicates that if inefficiency factors are fully addressed, output can be expanded by at least 47% on average while utilizing present resources and technology, or, to put it more precisely, if appropriate measures are taken to improve efficiency level, output can be expanded by up to 47% on average. Farmers still have the potential to improve their level of technical efficiency, as seen by the wide range of estimations of technical efficiency, which demonstrate that they continue to use resources inefficiently during the production process. According to this study, a substantial number of households were not achieving the best use of their production resources, which means they were not producing the most output possible from the available inputs.

**Table 4. Frequency distribution of technical efficiency of onion producers.**

| TE Level | Frequency | Percent |
|---|---|---|
| 0.05–0.20 | 16 | 7.80 |
| 0.20–0.40 | 47 | 22.93 |
| 0.40–0.60 | 54 | 26.34 |
| 0.60–0.80 | 78 | 38.05 |
| ≥0.80 | 10 | 4.88 |
| Total | 205 | 100 |
| Mean | 0.53 | |
| Minimum | 0.055 | |
| Maximum | 0.873 | |

Source: Computed from Field Survey Data, 2015/16

Fig 2 displays the frequency distribution of the predicted technical efficiencies to provide a clear picture of the distribution of the technical efficiencies. The majority of households have technical efficiencies that are between 0.60 and 0.80, according to the frequency of occurrences of the predicted technical efficiencies in the range. According to the sample frequency distribution, 38% of the respondents are concentrated in the technical efficiency ranges of 0.60 to 0.80. The results also show that the study areas' least and most technically efficient farmers are separated by a wide difference.

## 3.4. Yield gap due to technical inefficiency

The difference between technically efficient yield and actual yield is known as the yield gap. As a result, the yield gap is the amount that corresponds to lower yields as a result of technical inefficiency. The technical efficiency of the $i^{th}$ household is estimated using the stochastic

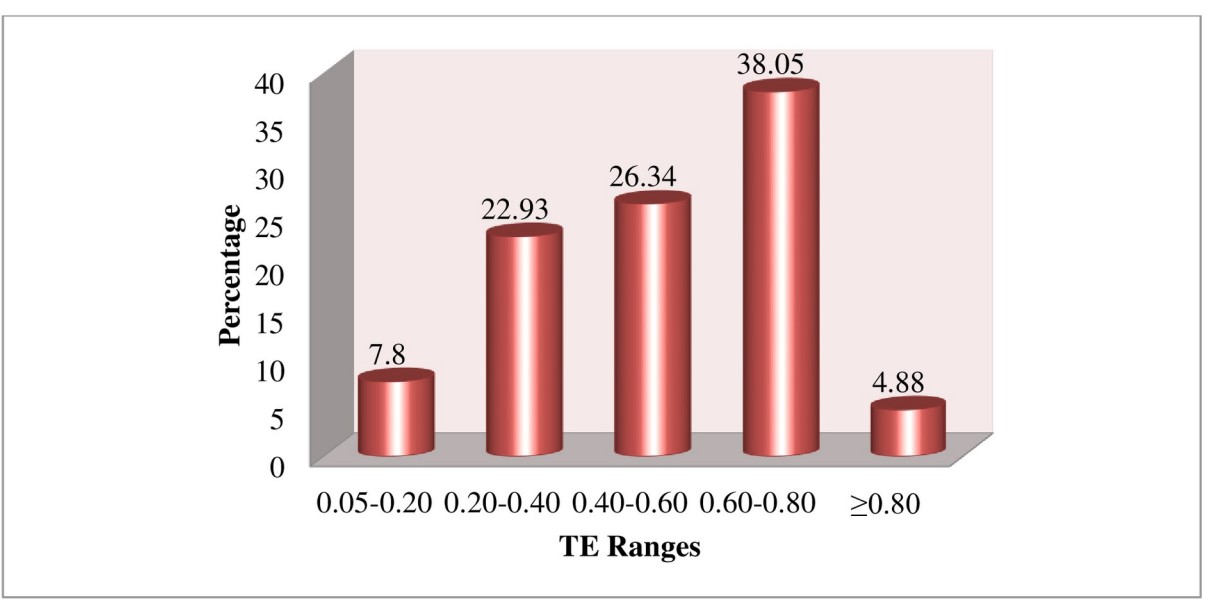

**Fig 2. Frequency distribution of technical efficiency.** Source: Computed from Field Survey Data, 2015/16.

**Table 5. Onion yield gap due to technical inefficiency.**

| Variable | Min | Max | Mean | Std. Dev. |
|---|---|---|---|---|
| Actual yield (kg/ha) | 200 | 25,000 | 2,965 | 2,958 |
| TE estimates | 0.055 | 0.873 | 0.53 | 0.20 |
| Potential/frontier yield (kg/ha) | 1,282.3 | 28,650.8 | 4,953 | 3,435 |
| Yield gap/loss (kg/ha) | 421.4 | 7,006.9 | 1,988 | 1,075.6 |

Source: Computed from Field Survey Data, 2015/16

model given in Eq (4) to be:

$$\text{TE}_i = \frac{Y_i}{Y_i^*} = \frac{f(X_i; \beta)\exp(v_i - \mu_i)}{f(X_i; \beta)\exp(v_i)} = \exp(-\mu_i)$$

Then, solving for $Y_i^*$, the potential yield of each household is represented as:

$$Y_i^* = \frac{Y_i}{\text{TE}_i} = f(X_i; \beta)\exp(v_i)$$

Where $\text{TE}_i$ = technical efficiency of the $i^{th}$ sample household in onion production
$Y_i^*$ = The frontier/potential output of the $i^{th}$ sample household in onion production, and
$Y_i$ = The actual/observed output of the $i^{th}$ sample household in onion production.

The potential onion output was calculated for each sample household in onion production on a hectare basis using the equation above, the values of the actual onion output obtained, and the predicted technical efficiency indices. Table 5 below shows the mean result.

It was found that the average yield gap for onions was 1,988 kg/ha, with the mean value of the actual output and the potential output being 2,965 kg/ha and 4,953 kg/ha, respectively. This indicates that the average technical inefficiency was 47%. This demonstrates that the average amount of onions produced in the study area by sample households was 1,988 kg/ha less than their potential yield.

During the production year, the mean values of both the actual and potential output were 2,965 kg/ha and 4,965 kg/ha with standard errors of 2,958 and 3,435, respectively in Table 5. According to the best-practice farms in the study area, Fig 3 shows that there is room to boost onion productivity under the prevailing practices.

## 4. Conclusion and recommendation

This research was conducted to estimate technical efficiency among small-scale irrigated onion-producing farmers and to identify the determinant factors for the technical efficiency of onion producers in the North Gondar Zone. The test result indicates that the traditional average response function is not an adequate representation of the production frontier and the decreasing returns to scale characteristics of the stochastic frontier production function. A significant proportion of the residual variation in the stochastic production frontier is due to technical inefficiency. This means that increased technical efficiency could be used to make improvements. The estimated Cobb-Douglas stochastic production frontier demonstrates the substantial inefficiency in onion production among plots. The output level can be enhanced by 47%, according to the mean efficiency level of 0.53. The degree of their efficiency varies significantly amongst plots as well. The increase in technical efficiency will therefore result in a sizable gain if inputs are used to the fullest extent possible. Out of six input variables, three input variables, which are oxen, plot size, and DAP, positively affected irrigated onion production.

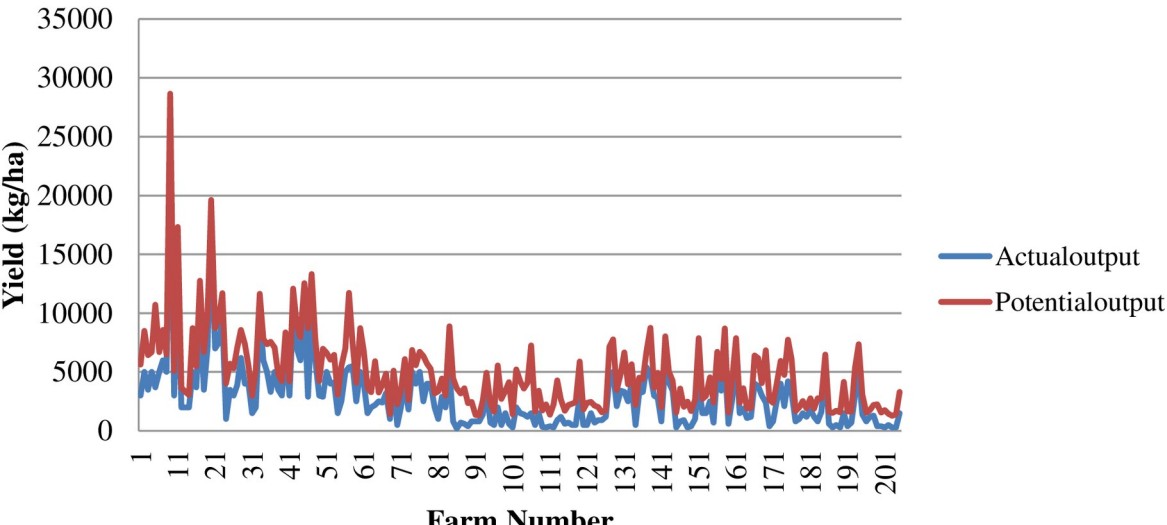

**Fig 3. Comparison of the actual and the potential level of yield.** Source: Computed from Field Survey Data, 2015/16.

The positive coefficient of these factors shows that increased use of these inputs will raise the production level by a greater amount. The estimated SPF model together with the inefficiency parameters shows the level of education, TLU, experience, and watering frequency influenced inefficiency negatively, whereas family size and training in marketing increased the level of technical inefficiency. Based on the findings, the subsequent recommendations are forwarded.

- It was found that education had a positive effect on technical efficiency, whereas training on onion marketing had a negative effect. Therefore, it is important to deliver continual scheduled training and education (integrated adult education) at the existing farmers' training centers (FTC) in order to empower the onion producers.

- The technical efficiency of producing irrigated onions is positively affected by livestock. Therefore, with the assistance of experts, modern livestock production techniques should be used.

- Onion production has benefited and been significantly affected by watering frequency. Furrow irrigation, which is currently used by farmers, is inefficient, so further groundwater resources and proper watering technologies should be used.

- Family size has a negative effect on the technical efficiency of producing irrigated onions. As a result, current family planning policies should be amended, and surplus labor should be used in other sectors.

- Experience has a positive effect on the technical efficiency of producing irrigated onions. As a result, more experienced onion producers ought to share their knowledge with less experienced ones.

## Supporting information

**S1 Dataset.**
(XLSX)

**S1 File. Questionnaire survey.**
(DOCX)

**S1 Appendix.**
(DOCX)

## Acknowledgments

We presented thanks to the districts (Dembia, Gondar Zurai, and Takusa) for facilitating data collections and providing data; the farmers who provided the primary data.

## Author Contributions

**Conceptualization:** Tigabu Dagnew Koye, Abebe Dagnew Koye, Zework Aklilu Amsalu.

**Data curation:** Tigabu Dagnew Koye, Abebe Dagnew Koye.

**Formal analysis:** Tigabu Dagnew Koye, Zework Aklilu Amsalu.

**Funding acquisition:** Tigabu Dagnew Koye, Abebe Dagnew Koye.

**Investigation:** Tigabu Dagnew Koye, Abebe Dagnew Koye, Zework Aklilu Amsalu.

**Methodology:** Tigabu Dagnew Koye.

**Project administration:** Tigabu Dagnew Koye.

**Resources:** Tigabu Dagnew Koye, Abebe Dagnew Koye, Zework Aklilu Amsalu.

**Software:** Tigabu Dagnew Koye, Abebe Dagnew Koye, Zework Aklilu Amsalu.

**Supervision:** Tigabu Dagnew Koye, Abebe Dagnew Koye, Zework Aklilu Amsalu.

**Validation:** Tigabu Dagnew Koye, Abebe Dagnew Koye, Zework Aklilu Amsalu.

**Visualization:** Tigabu Dagnew Koye, Abebe Dagnew Koye, Zework Aklilu Amsalu.

**Writing – original draft:** Tigabu Dagnew Koye, Abebe Dagnew Koye.

**Writing – review & editing:** Tigabu Dagnew Koye, Abebe Dagnew Koye, Zework Aklilu Amsalu.

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
