## [Decision Letter · Decision Letter 0]

26 Apr 2022

PONE-D-22-07294Analysis of Technical Efficiency of Irrigated Onion (Allium cepa L.) Production in North Gondar Zone of Amhara Regional State, EthiopiaPLOS ONE

Dear Dr. Koye,

Thank you for submitting your manuscript to PLOS ONE. After careful consideration, we feel that it has merit but does not fully meet PLOS ONE’s publication criteria as it currently stands. Therefore, we invite you to submit a revised version of the manuscript that addresses the points raised during the review process.

We look forward to receiving your revised manuscript.

Kind regards,

Vassilis G. Aschonitis

Academic Editor

PLOS ONE

Journal Requirements:

3. Thank you for submitting the above manuscript to PLOS ONE. During our internal evaluation of the manuscript, we found significant text overlap between your submission and the following previously published works, some of which you are an author.

- https://www.iiste.org/Journals/index.php/JBAH/article/viewFile/31054/31886

- http://etd.aau.edu.et/bitstream/handle/123456789/8858/Girma%20Masresha.pdf

- https://www.omicsonline.org/open-access/growth-and-bulb-yield-of-onion-allium-cepa-l-in-response-to-plant-density-and-variety-in-jimma-south-western-ethiopia-2329-8863-1000357-100893.html

- https://assets.researchsquare.com/files/rs-150323/v1_covered.pdf?c=1631851944

- https://journalofeconomicstructures.springeropen.com/articles/10.1186/s40008-019-0166-y

- file:///C:/Users/canderson/Downloads/109844-Article%20Text-301838-1-10-20141118%20(1).pdf

- http://ijrar.com/upload_issue/ijrar_issue_20543549.pdf

-http://unimaid.edu.ng/Journals/Agriculture/JAEESS%20-%20Agric%20Econs/11.pdf

- http://iresearcher.org/IR%20Template,%20V9%20I1%202020%20-Jemil.pdf

Please revise the manuscript to rephrase the duplicated text, cite your sources, and provide details as to how the current manuscript advances on previous work. Please note that further consideration is dependent on the submission of a manuscript that addresses these concerns about the overlap in text with published work.

4. Please include additional information regarding the survey or questionnaire used in the study and ensure that you have provided sufficient details that others could replicate the analyses. For instance, if you developed a questionnaire as part of this study and it is not under a copyright more restrictive than CC-BY, please include a copy, in both the original language and English, as Supporting Information.

6. Thank you for stating the following financial disclosure: 

"This study has been funded by the University of Gondar"

7. Thank you for stating the following in the Acknowledgments Section of your manuscript: 

"the University of Gondar was given financial support for the successful completion of this research work"

"the University of Gondar was given financial support for the successful completion of this research work"

8. In your Data Availability statement, you have not specified where the minimal data set underlying the results described in your manuscript can be found. PLOS defines a study's minimal data set as the underlying data used to reach the conclusions drawn in the manuscript and any additional data required to replicate the reported study findings in their entirety. All PLOS journals require that the minimal data set be made fully available. For more information about our data policy, please see http://journals.plos.org/plosone/s/data-availability.

9. PLOS requires an ORCID iD for the corresponding author in Editorial Manager on papers submitted after December 6th, 2016. Please ensure that you have an ORCID iD and that it is validated in Editorial Manager. To do this, go to ‘Update my Information’ (in the upper left-hand corner of the main menu), and click on the Fetch/Validate link next to the ORCID field. This will take you to the ORCID site and allow you to create a new iD or authenticate a pre-existing iD in Editorial Manager. Please see the following video for instructions on linking an ORCID iD to your Editorial Manager account: https://www.youtube.com/watch?v=_xcclfuvtxQ

Reviewers' comments:

Reviewer's Responses to Questions

**Comments to the Author**

1. Is the manuscript technically sound, and do the data support the conclusions?

Reviewer #1: Yes

2. Has the statistical analysis been performed appropriately and rigorously? 

Reviewer #1: Yes

3. Have the authors made all data underlying the findings in their manuscript fully available?

Reviewer #1: Yes

4. Is the manuscript presented in an intelligible fashion and written in standard English?

Reviewer #1: Yes

5. Review Comments to the Author

Reviewer #1: I have read your manuscript and I think the experiments and the data presentation are sound.

1. The manuscript didn’t have line number.

2. What’s the hypothesis of this paper?

3. Minor suggestions sees the attached file.

6. PLOS authors have the option to publish the peer review history of their article (what does this mean?). If published, this will include your full peer review and any attached files.

Reviewer #1: No

---

## [Author Response · Author response to Decision Letter 0]

5 Sep 2022

Dear editors: I have tried to revise it based on the given comments. So, please see the revised manuscript.

Authors’ response

For Reviewer 1:

IN ABSTRACT PART

“Onion plays an important role in contributing to a household’s food security and income. But, limited improved seeds, low fertilizers application, low marketing infrastructure, weak institutional support, and market access have contributed to onion’s low productivity”. This problem is a regional issue.

Authors’ response: It will be specify the problems for onion production i.e. “Onion plays an important role in contributing to a household’s food security and income. Despite its importance in the human diet, economic profit and its increasing area coverage, the productivity of onion in the country is much lower on average value.”

Show the full name of DAP?

Authors’ response: Di Ammonium Phosphate

INTRODUCTION PART

The recent data need to be shown (in paragraph one)

Authors’ response: it will be changed based on the recent data “Onion (Allium Cepa) is a vital vegetable crop that’s a complimentary product to tomatoes and is of major commercial importance throughout the planet (Przygocka-Cyna et al., 2020). Worldwide production of onion in the year 2019 was 99,968,016 Mg which makes it second to tomato and accounted for 9% of the total share of vegetables (‘World Food Agric. – Stat. Yearb. 2021,’ 2021). China is the leading onion producer with 23,907,509 tons, followed by India 19,415,425 Mg, Egypt 3,115,482 Mg, United States of America 3,025,700 Mg, Iran 2,345,768 Mg and Turkey 2,120,581 Mg (Bornhofen et al., 2019; Hanci, 2018)”.

What’s the hypothesis of this paper?

Authors’ response: the working hypothesis (null hypothesis) sated as follows:

• Smallholder farmers are not technically efficient in irrigated onion production in the study area

• The socioeconomic variables do not significantly influence technical inefficiency

MATERIALS AND METHODS

Description of the Study Area: it is better to present a map of study area

Authors’ response: According to the reviewer’s comment, we have built the map of the study area. Therefore, the map is put in the revised manuscript i. e. on figure 1.

RESULTS AND DISCUSSIONS 

Descriptive analysis: what’s the unit of this value (the standard deviation?)

Authors’ response: the standard deviation is unit-less.

The manuscript didn’t have line number

Authors’ response: most of the line numbers given by journals. if it is necessary, I will give line numbers in the next.

---

## [Decision Letter · Decision Letter 1]

12 Sep 2022

Analysis of Technical Efficiency of Irrigated Onion ( Allium cepa L. ) Production in North Gondar Zone of Amhara Regional State , Ethiopia

PONE-D-22-07294R1

Dear Dr. Koye,

We’re pleased to inform you that your manuscript has been judged scientifically suitable for publication and will be formally accepted for publication once it meets all outstanding technical requirements.

Kind regards,

Vassilis G. Aschonitis

Academic Editor

PLOS ONE

Additional Editor Comments (optional):

Reviewers' comments:

Reviewer's Responses to Questions

**Comments to the Author**

1. If the authors have adequately addressed your comments raised in a previous round of review and you feel that this manuscript is now acceptable for publication, you may indicate that here to bypass the “Comments to the Author” section, enter your conflict of interest statement in the “Confidential to Editor” section, and submit your "Accept" recommendation.

Reviewer #1: All comments have been addressed

2. Is the manuscript technically sound, and do the data support the conclusions?

Reviewer #1: Yes

3. Has the statistical analysis been performed appropriately and rigorously? 

Reviewer #1: Yes

4. Have the authors made all data underlying the findings in their manuscript fully available?

Reviewer #1: Yes

5. Is the manuscript presented in an intelligible fashion and written in standard English?

Reviewer #1: Yes

6. Review Comments to the Author

Reviewer #1: authors have revised the manuscript according to comments of reviewers, it could be accepted for publication in the journal.

7. PLOS authors have the option to publish the peer review history of their article (what does this mean?). If published, this will include your full peer review and any attached files.

Reviewer #1: No

---

## [Editor Report · Acceptance letter]

26 Sep 2022

PONE-D-22-07294R1 

Analysis of Technical Efficiency of Irrigated Onion (*Allium cepa* L.) Production in North Gondar Zone of Amhara Regional State, Ethiopia 

Dear Dr. Koye:

I'm pleased to inform you that your manuscript has been deemed suitable for publication in PLOS ONE. Congratulations! Your manuscript is now with our production department. 

Kind regards, 

on behalf of

Dr. Vassilis G. Aschonitis 

Academic Editor

PLOS ONE